# Effects of organic fertilizer on soil nutrient status, enzyme activity, and bacterial community diversity in *Leymus chinensis* steppe in Inner Mongolia, China

**Lirong Shang, Liqiang Wan** **\*, Xiaoxin Zhou, Shuo Li, Xianglin Li\***

Institute of Animal Sciences, Chinese Academy of Agricultural Sciences, Beijing, China

\* wanliqiang@caas.cn (LW); lxl@caas.cn (XL)

## Abstract

The long-term impact of human exploitation and environmental changes has led to a decline in grassland productivity and soil fertility, which eventually results in grassland degradation. The application of organic fertilizer is an effective improvement measure; however, it is still not fully understood how the addition of organic fertilizer influences grassland soil fertility and plant composition. A set of experiments were conducted in Inner Mongolia in China to reveal the tradeoff between steppe plants and soil microorganisms and the eco-physiological mechanisms involved, and how the addition of vermicompost and mushroom residues affect microbial diversity, enzyme activities, and the chemical properties of soil in degraded *Leymus chinensis* grassland. Organic fertilizer improved the soil nutrient status and shaped distinct bacterial communities. Compared with the control the available phosphorus (AP) and available potassium (AK) contents were highest under treatments a3 and b3, and the aboveground biomass was highest under the b3 treatment. Soil sucrase activities increased by 7.88% under the b3 treatment. Moreover, the richness index significantly increased by 7.07% and 7.23% under the a1 and b2 treatments, respectively. The most abundant Actinobacteria and Proteobacteria were detected in the organic fertilizer treatment. A linear discriminant analysis effect size (LEfSe) indicated that the bacterial community was significantly increased under the b3 treatment. A canonical correspondence analysis (RDA) and spearman correlation heatmap confirmed that total P (TP) and urease were the key driving factors for shaping bacterial communities in the soil. Our results indicated that the application of large amounts of vermicompost and mushroom residues increased the availability of nutrients and also enhanced the biodiversity of soil bacterial communities in *L. chinensis* grasslands, which will contribute to the sustainable development of agro-ecosystems.

## Introduction

*Leymus chinensis* grassland forms the main pastoral area and animal husbandry production base in the Hulunbeier steppe region [1]. Due to the long-term use of the grassland, particularly for grazing, the large systematic nutrient output has led to widespread serious grassland

**Funding:** This work was supported by the National Key Research and Development Program (2016YFC0500608-2) and the National Forage Industry Technology System (CARS-34).

**Competing interests:** The authors have declared that no competing interests exist.

degeneration [2, 3]. Land degradation is an important indicator of ecosystem degradation [4], and soil fertility decline is considered to be a primary cause of the low productivity of grasslands in northern China [5]. Fertilizer application is extensively used as a common management practice to maintain soil fertility and crop productivity [6]. A reasonable fertilizer application can improve the yield and quality of pasture, change soil pH, and improve other aspects of the soil [7, 8]; thus, affecting the soil microbial activity and nutrient conversion process [9, 10]. In the pursuit of economic growth and food production, increasing amounts of chemical fertilizers have been applied in agroecosystems worldwide [11, 12], which has resulted in serious physicochemical degradation of soil and a deterioration in productivity [13, 14]. Organic fertilizers are derived from animal or plant matter, and their application can modify soil physicochemical conditions due to the abundance of organic matter and balancing of nutrient levels [15–17]. Many studies have shown that organic amendments are an effective way to improve soil fertility, crop yield, and environmental quality [18–20]. The vermicompost and mushroom residue organic fertilizers used in this study were obtained from the livestock manure of a local farm and the substrate remaining after planting agricultural by-products, to achieve the recycling of nutrients.

Soil microorganisms are an important part of grassland ecosystems and play a leading role in material transformation, energy flows, and organic matter degradation [2]. As the most important and active part of the soil ecosystem, they are involved in almost all soil life processes. Because of the sensitivity of soil quality to the soil microbial population, many studies have used soil microbial parameters as a measure of soil quality, especially in the evaluation of soil quality after fertilizer application [21, 22]. Soil microbial community composition, microbial diversity, microbial respiration, and soil enzymes that respond to microbial activity can be used to determine changes in soil quality [23]. Due to worldwide concerns regarding the restoration of grassland degradation in recent years, the number of study on the effects of fertilizer application on soil microbial in grassland have gradually increased. Examples include studies of the soil microbial community and soil quality following fertilizer application [24], the effects of different proportions of organic and inorganic fertilizers on soil [25], and the effects of fertilizer application on soil organic carbon (SOC) composition and microbial community structure [26]. Therefore, investigations of soil microbial communities are vital for understanding the interactions among soil, microbes, and their host plants following the application of organic fertilizers, which will enable the use of more effective fertilizer application regimes [27].

In this study, a field experiment to test the application of vermicompost and mushroom residue organic fertilizers was conducted on *L. chinensis* grassland in the Hulunbuir steppe region was conducted. The influence of different organic fertilizer amounts on the soil bacterial community structure and composition in natural grassland in Inner Mongolia was evaluated. The objectives were: (i) to identify the effects of different organic fertilizers and fertilizer application amounts on soil chemical properties and enzyme activities, (ii) to determine the effect of different organic fertilizers on soil bacterial diversity and richness; (iii) to explore the relative abundances of the dominant bacterial phyla under different organic fertilizers and fertilizer application amounts; and (iv) to clarify the optimum amount of two organic fertilizers in degraded *L. chinensis* steppe.

## Materials and methods

### Study area

This study was conducted at the Hulunbuir Grassland Ecosystem Observation and Research Station located at Xiertala farm in the center of the Hulunbuir meadow steppe (49°25′N, 119°

70′E) in the north-eastern region of Inner Mongolia, China. The climate zone is continental temperate semi-arid. The soil of the study site is a chestnut soil, elevation is 649.6 m, the highest temperature is 36.2°C, the lowest temperature is –48.5°C, the annual average temperature is –2.4°C, the $\geq$10°C annual accumulated temperature is 1,500–1,800°C, the frost-free period is about 110 d; annual average rainfall is 350–400 mm (mostly concentrated in July–September), and rain and heat occur in the same period. The vegetation type is a typical grassland with *L. chinensis* as the main species, but other dominant species include *Stipa baicalensis roshev*, *Cleistogenes squarrosa*, etc., accompanied by *Poa pratensis*, *Thalictrum squarrosum*, and *Carex duriuscula* C.A.

The experiment included seven fertilization treatments: (1) unfertilized control (ck), (2) vermicompost 15 t hm$^{-2}$ (a1), (3) vermicompost 30 t hm$^{-2}$ (a2), (4) vermicompost 45 t hm$^{-2}$ (a3), (5) mushroom residue 15 t hm$^{-2}$ (b1), (6) mushroom residue 30 t hm$^{-2}$ (b2), and (7) mushroom residue 45 t hm$^{-2}$ (b3). The field experiment was conducted based on a completely single factor randomized design, with three replications of each treatment, i.e., a total of 21 plots. Each plot had an area of 15 m$^2$ (3 × 5 m), with an intermediate interval of 1 m. The vermicompost used in this test was mainly cow dung, with the following nutrient content: soil organic matter (SOM) 26.08%, total nitrogen (TN) 10.65 g kg$^{-1}$, available phosphorus (AP) 282.23 mg kg$^{-1}$, available potassium (AK) 2839.20 mg kg$^{-1}$, and water content (WC) 0.98%. The mushroom residue was mainly the substrate after mushrooms were planted, with the following nutrient content: SOM 24.61%, TN 12.39 g kg$^{-1}$, AP 382.01 mg kg$^{-1}$, AK 690.70 mg kg$^{-1}$, and WC 1.01%. Two organic fertilizers were provided by the National Field Science Observatory of the Hulunbuir Grassland Ecosystem of the CAAS and were artificially applied to the surface in mid-June 2018. Soil was collected from the 0–20 cm layer early in August 2018. All samples were sealed in sterile plastic bags, packed on ice, and transported to the laboratory immediately, where they were sieved through 2 mm meshes and thoroughly homogenized after removing plant residues and gravel. Each sample was then divided into three parts: one sub-sample was air-dried for the analysis of soil chemical properties, one sub-sample was stored at 4°C for enzyme analysis, and the third sub-sample was placed in a centrifuge tube for the determination of microorganisms.

**Soil chemical properties and biomass.** The available N (AN, alkalized N method), AP (0.5 M NaHCO$_3$ extraction), and AK (1.0 M ammonium acetate extraction) were determined following the procedures described by Lu [28]. Soil TN was determined by the semimicro-Kjeldahl method. The TP and TK were extracted and determined by the perchloric acid digestion method and spectrophotometry protocols [29, 30]. The SOM was measured by the K$_2$Cr$_2$O$_7$ external heating method. Biomass was determined using a weighing method.

## Enzyme activity analysis

Soil urease activity was determined by indophenol colorimetry and expressed by the mass (μg) of NH$_3$-N produced in 1 g of soil after a 24 h incubation at 37°C, and activity was measured at a wavelength of 578 nm using a spectrophotometer.

Sucrase activity was determined by the 3,5-dinitrosalicylic acid colorimetric method, and expressed as the mass (mg) of glucose released in 1 g of soil incubated at 37°C for 24 h. It was measured at a wavelength of 510 nm using a spectrophotometer.

Alkaline phosphatase activity was determined by phenyldisodium phosphate colorimetry and activity was expressed as the number of μmol of phenol in 1 g of soil after 24 h. It was measured at a wavelength of 578 nm using a spectrophotometer.

Catalase activity was determined by the 0.3% $H_2O_2$ colorimetric method, and expressed as the hydrogen peroxide (μmol) decomposed by 1 g of soil in 24 h. It was measured at a wavelength of 240 nm [31].

## DNA extraction, Polymerase Chain Reaction (PCR), and sequencing

Microbial community genomic DNA was extracted from each sample using the PowerSoil DNA Isolation Kit (Mo Bio Laboratories, Solana Beach, CA, USA) according to the manufacturer's instructions.

The DNA extract was placed on 1% agarose gel, and the DNA concentration and purity were determined with a NanoDrop 2000 UV-vis spectrophotometer (Thermo Fisher Scientific, Waltham, MA, USA). The hypervariable region V3-V4 of the bacterial 16S rRNA gene were amplified with the primer pairs 338F (5′-ACTCCTACGGGAGGCAGCAG-3′) and 806R (5′-GGACTACHVGGGTWTCTAAT-3′) by an ABI GeneAmp® 9700 PCR thermocycler (ABI, Foster City, CA, USA). The PCR amplification of the 16S rRNA gene was performed as follows: initial denaturation at 95°C for 3 min, followed by 27 cycles of denaturing at 95°C for 30 s, annealing at 55°C for 30 s, an extension at 72°C for 45 s, and single extension at 72°C for 10 min, with an end at 4°C. The PCR mixtures contained 5 × TransStart FastPfu buffer 4 n at of denaturing at saturation at 9er (5 μM) 0.8 μL, reverse primer (5 μM) 0.8 μL, TransStart FastPfu DNA Polymerase 0.4 μL, template DNA 10 ng, and finally ddH$_2$O up to 20 μL. The PCR reactions were performed in triplicate. The PCR product was extracted from 2% agarose gel and purified using an AxyPrep DNA Gel Extraction Kit (Axygen Biosciences, Union City, CA, USA) according to the manufacturer's instructions and quantified using a Quantus™ Fluorometer (Promega, San Luis Obispo, CA, USA). Purified amplicons were pooled in equimolar and paired-end sequences (2 × 300) on an Illumina MiSeq platform (Illumina, San Diego, CA, USA) according to the standard protocols reported by Majorbio Bio-Pharm Technology Co. Ltd. (Shanghai, China). The raw reads were deposited into the National Center for Biotechnology Information (NCBI) Sequence Read Archive (SRA) database.

## Processing of sequencing data

The raw 16S rRNA gene sequencing reads were demultiplexed, quality-filtered by Trimmomatic and merged by FLASH, with the following criteria. (i) The 300 bp reads were truncated at any site receiving an average quality score of <20 over a 50 bp sliding window, and truncated reads shorter than 50 bp were discarded. Reads containing ambiguous characters were also discarded. (ii) Only overlapping sequences longer than 10 bp were assembled according to their overlapped sequence. The maximum mismatch ratio of the overlap region was 0.2. Reads that could not be assembled were discarded. (iii) Samples were distinguished according to the barcode and primers, and the sequence direction was adjusted. Exact barcodes were matched, with a maximum of two nucleotide mismatches per primer.

Operational taxonomic units (OTUs) with a 97% similarity cutoff were clustered using UPARSE (version 7.1, http://drive5.com/uparse/), and chimeric sequences were identified and removed. The taxonomy of each OTU representative sequence was analyzed by the RDP Classifier (http://rdp.cme.msu.edu/) against the 16S rRNA database (Silva SSU128) using a confidence threshold of 0.7.

## Statistical analysis

Soil chemical properties, enzyme activity, and alpha diversity (Shannon, Ace, and coverage indexes) were subjected to an analysis of variance (ANOVA), performed using IBM SPSS 20.0 (SPSS Inc., USA). A one-way ANOVA was used to compare the significance of differences

between different treatments. Species richness and biodiversity were estimated by the Ace estimator (Ace), Shannon diversity index (Shannon), and Good's nonparametric coverage (Coverage). Values of $P < 0.05$ were considered statistically significant. A partial least squares—discriminant analysis (PLS-DA) was applied to analyze the differences in bacterial community structure among soil samples. The relationship between soil microbial community structure and each affecting factor was analyzed by a redundancy analysis (RDA) and variation partitioning. The RDA eliminated redundant variables depending on other measured variables, automatically selecting variables with large effects. Based on the Kruskal-Wallis (KW) sum-rank test, a linear discriminant analysis (LDA) effect size (LEfSe) was performed to identify significantly different species of bacterial taxa among groups, and a threshold LDA score $\geq 2.0$ was considered to be an important contributor to the model. All of these analyses were completed on the I-sanger platform (Majorbio, Shanghai, China) based on various packages in R (www.R.project.org) and the Galaxy web application and workflow framework (http://huttenhower.sph.harvard.edu/galaxy/).

## Results

### Soil chemical properties

The application of vermicompost and mushroom residue organic fertilizers significantly altered soil chemical properties (Table 1). The vermicompost and mushroom residue applications led to a significant increase in the soil AP and AK ($P < 0.05$). In addition, the AP and AK were highest under the a3 and b3 treatments, respectively. Soil AN and SOM were not significantly affected by vermicompost and mushroom residue applications. Soil TN, TP and TK also didn't show significant variations. Both total and available nutrients were enhanced by the vermicompost and mushroom residue treatments, indicating an effect of vermicompost and mushroom residues on chemical properties. Furthermore, aboveground biomass was significantly higher under the b3 treatment than for other treatments.

### Soil enzyme activities

Four types of soil enzymes (urease, sucrase, alkaline phosphatase, and catalase) did not significantly respond to the addition of vermicompost and mushroom residue fertilizer (Table 2). Urease activities declined as a result of vermicompost and mushroom residue applications. However, sucrase, alkaline phosphatase, and catalase activities increased to different degrees under the different levels of treatment of vermicompost and mushroom residue fertilizers.

**Table 1. Variations in soil chemical properties under different vermicompost and mushroom residue fertilizer application.**

| Treatment | TN (g kg⁻¹) | TP (g kg⁻¹) | TK (g kg⁻¹) | AN (mg kg⁻¹) | AP (mg kg⁻¹) | AK (mg kg⁻¹) | SOM (%) | DM (g m⁻²) |
|---|---|---|---|---|---|---|---|---|
| ck | 2.51±0.25a | 0.66±0.12a | 31.85±0.61a | 172.84±10.24a | 7.43±0.94b | 179.42±31.01c | 5.55±0.09a | 172.8±10.68c |
| a1 | 2.52±0.30a | 0.68±0.10a | 32.64±1.44a | 175.90±11.67a | 8.89±1.39b | 209.19±55.79bc | 5.22±0.62a | 198.38±14.94bc |
| a2 | 2.26±0.12a | 0.69±0.21a | 31.96±0.84a | 167.76±3.12a | 8.34±1.29b | 195.03±32.10bc | 5.03±0.52a | 180.91±2.90bc |
| a3 | 2.47±0.08a | 0.73±0.20a | 33.12±0.71a | 184.54±15.22a | 10.20±0.15a | 252.14±4.68ab | 5.54±0.55a | 214.27±5.26bc |
| b1 | 2.56±0.28a | 0.74±0.26a | 32.18±0.51a | 183.70±3.66a | 9.06±1.87b | 250.67±12.13ab | 5.58±0.18a | 191.49±1.99ab |
| b2 | 2.10±0.07a | 0.58±0.04a | 31.84±0.56a | 169.34±6.80a | 8.18±0.48b | 220.20±23.32bc | 4.83±0.56a | 219.78±10.73ab |
| b3 | 2.40±0.26a | 0.55±0.01a | 32.59±0.88a | 187.96±5.60a | 8.84±0.31b | 292.44±20.67a | 5.58±0.44a | 244.11±13.59a |

Date are presented as the mean ± standard variance (STDEV, n = 3). Different letters within columns indicate significance at $P < 0.05$ according to Duncan's test. ck, no fertilization treatment; a1, a2, a3, vermicompost fertilizer treatments; b1, b2, b3, mushroom residue fertilizer treatments. TN, total nitrogen; TP, total phosphorus; TK, total potassium; AN, available nitrogen; AP, available phosphorus; AK, available potassium; SOM, soil organic matter.

**Table 2. Variations in soil enzyme activities under different vermicompost and mushroom residue fertilizer application.**

| Treatment | S-UE | S-SC | S-AKP | S-CAT |
|---|---|---|---|---|
| ck | 805.61±94.01a | 106.15±9.05a | 13.20±2.27a | 45.90±2.54a |
| a1 | 768.44±31.33a | 111.52±4.57a | 12.85±2.33a | 44.39±2.40a |
| a2 | 716.61±85.57a | 104.24±7.97a | 12.25±0.94a | 43.74±2.47a |
| a3 | 673.32±52.25a | 103.97±5.38a | 13.99±1.00a | 48.04±2.52a |
| b1 | 700.21±111.33a | 107.71±6.21a | 14.48±2.30a | 45.98±2.70a |
| b2 | 740.23±72.37a | 112.81±2.84a | 11.32±1.40a | 44.01±2.38a |
| b3 | 704.59±51.84a | 114.51±0.75a | 13.11±2.26a | 45.14±2.40a |

S-UE, soil Urease; S-SC, soil sucrase; S-AKP, soil alkaline phosphatase; S-CAT, soil catalase. Data are presented as the mean ± standard variance (STDEV, n = 3). Different letters within columns indicate significance at P < 0.05 according to Duncan's test. ck, no fertilization treatment; a1, a2, a3, vermicompost fertilizer treatments; b1, b2, b3, mushroom residue fertilizer treatments.

Compared with ck treatment, soil sucrase activities increased by 7.88% under the b3 treatment. Similarly, soil alkaline phosphatase and catalase activities increased by 6.01% and 4.65% under the a3 treatment, respectively.

## General analyses of the sequencing data

A total of 1405452 sequence reads were successfully elicited from all soil samples. After removing short and low-quality reads, singletons, replicates, and chimeras, 955201 sequences, ranging from 35708 to 53884 per sample, were retained. Based on 97% similarity, a total of 3481 OTUs, ranging from 2193 to 2506 per sample, were obtained across all samples (S1 Table). Among the total sequences, ~99.4% were classified as bacteria, with 31 phyla, 75 classes, 153 orders, and 298 families.

A rarefaction analysis showed that the number of OTUs observed for the 16S rRNA gene reached saturation (S1 Fig), which indicated that the sequencing capability was large enough to capture the complete biodiversity of these communities.

## Richness and biodiversity of the bacterial community in the soil following vermicompost and mushroom residue treatments

The coverage indices of all treatments were greater than 0.98, which indicated that the sequencing capability was large enough to capture most of the bacterial community characteristics of each treatment (Table 3). The number of observed OTUs in the b2 treatment was significantly (P < 0.1) increased by 7.19% compared to the ck treatment. The a1 treatment had the highest Shannon diversity index (6.53), followed by the b2 treatment (6.51). There was no significant difference between the vermicompost, mushroom residue, and ck treatments. The richness index (Ace) in the ck treatment was significantly (P < 0.1) decreased by 7.07% and 7.23% compared to the a1 and b2 treatments, respectively.

## Soil bacterial microbial communities in the different organic fertilizer treatments

The PLS-DA revealed that the soil bacterial communities varied among the different fertilizer application treatments (Fig 1). Distinctly different clusters of bacterial communities were formed under the different levels of vermicompost and mushroom residue fertilizer treatments. In addition, the a1, a2, b3, and other treatments were clearly separated and distributed

**Table 3. Estimated number of observed OTUs, biodiversity, richness, and coverage across treatments.**

| Treatment | Observed OTUs | Shannon | Ace | Coverage |
| --- | --- | --- | --- | --- |
| ck | 2272.67±52.80b | 6.43±0.06a | 2703.45±53.03b | 0.9849±0.0002a |
| a1 | 2411.00±15.80ab | 6.53±0.01a | 2894.58±19.41a | 0.9837±0.0002a |
| a2 | 2396.00±39.38ab | 6.47±0.03a | 2848.59±58.42ab | 0.9841±0.0009a |
| a3 | 2301.67±18.31ab | 6.39±0.02a | 2755.35±23.34ab | 0.9843±0.0003a |
| b1 | 2303.67±72.76ab | 6.38±0.07a | 2791.93±88.16ab | 0.9838±0.0009a |
| b2 | 2436.00±1.70a | 6.51±0.02a | 2898.92±18.63a | 0.9838±0.0004a |
| b3 | 2372.00±54.95ab | 6.45±0.06a | 2839.53±76.17ab | 0.9839±0.0011a |

Data are presented as the mean ± SE (n = 3). Different letters within columns indicate significance at P < 0.1 according to Duncan's test. ck, no fertilization treatment; a1, a2, a3, vermicompost fertilizer treatments; b1, b2, b3, mushroom residue fertilizer treatments. Observed OTUs, observed operational taxonomic units; Shannon, nonparametric Shannon diversity index; Ace, richness of the Ace estimator; Coverage, Good's nonparametric coverage estimator.

to the left and right along the comp1. However, there was no clear difference between the a3 and ck treatments, and the community composition of the a2 and b3 treatments was similar.

## The effects of different organic fertilizers on bacterial community composition in the soil

The five most dominant phyla across all samples were *Actinobacteria*, *Proteobacteria*, *Acidobacteria*, *Verrucomicrobia*, and *Chlorofexi*, which together accounted for more than 85% of the relative abundance of the bacterial communities (Fig 2). Among these dominant phyla, *Actinobacteria* was most abundant in the a2 treatment (36.79%) but least abundant in the a1 treatment (31.57%). Conversely, *Acidobacteria* was most abundant in the a1 treatment (20.69%) and least abundant in the a2 treatment (13.92%). *Proteobacteria*, as the second-most abundant phylum, was most abundant in the b1 and b3 treatments (23.29% and 22.32%, respectively). *Verrucomicrobi*a were most abundant in the a3 treatment (12.72%) and *Chlorofexi* was most abundant in the b2 treatment (10.59%).

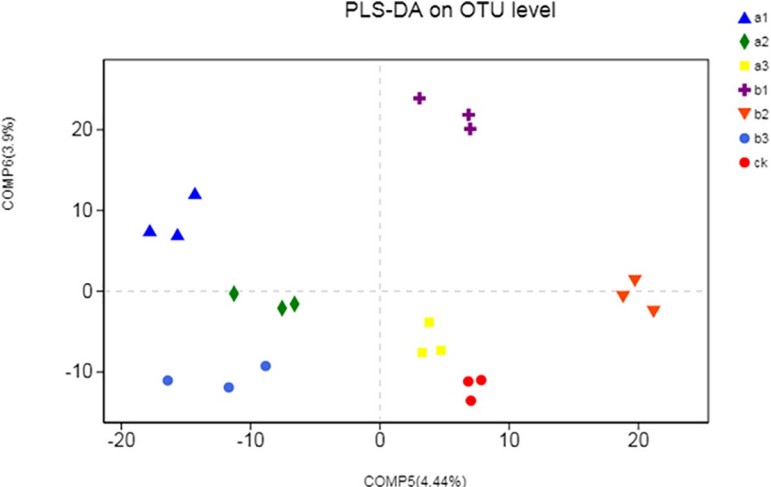

**Fig 1. A Partial Least Square—Discriminant Analysis (PLS-DA) score plot.** The PLS-DA is distinguishing the soil bacterial microbial communities of different organic fertilizer treatments. Note: ck, no fertilizer application treatment; a1, a2, a3, vermicompost fertilizer treatments; b1, b2, b3, mushroom residue fertilizer treatments.

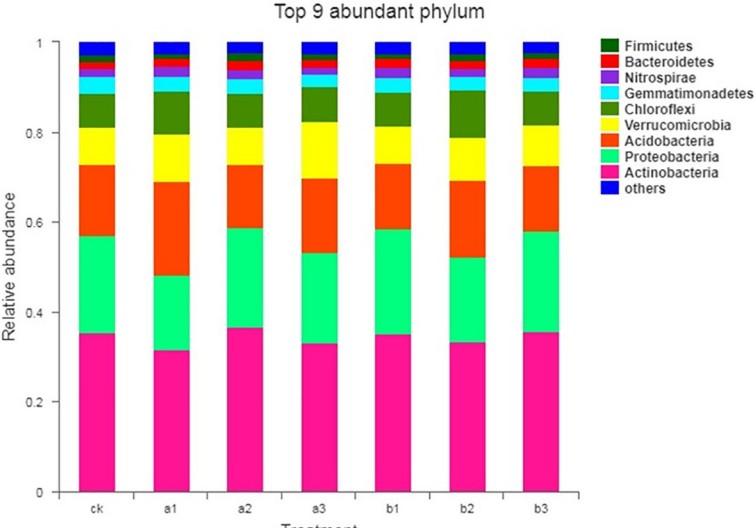

**Fig 2. The relative abundances of the top nine bacterial phyla present in the different treatments.** Values are the means of triplicate samples from each each treatment. Note: ck, no fertilizer application treatment; a1, a2, a3, vermicompost fertilizers treatments; b1, b2, b3, mushroom residue fertilizer treatments.

## Bacterial microbial communities with statistically significant differences

To identify the specific bacterial taxa associated with different organic fertilizer levels, we compared the bacterial communities in the ck, a1, a2, a3, b1, b2, and b3 treatments using LEfSe. Each circle at a different classification level in the evolution map represents a classification at that level. Yellow indicates no significant change in abundance and the diameter of the circle indicates relative abundance. This tool allows the analysis of microbial community data for any clade, but the results were only statistically analyzed from the phylum to the genus level (Fig 3). As shown in Fig 3A, 26 bacterial clades presented statistically significant differences, with LDA scores of 2 or higher. According to the LEfSe, more bacterial taxa (17 clades, 1 class, 1 order, 4 families, and 11 genera) were detected by LEfSe in the b3 treatment than in the other treatments, namely *Limnochordia* (class to family), *Pseudomonadaceae* (family to genus), *Luteimonas* (genus), *Pusillimonas* (genus), *Devosia* (genus), *Filomicrobium* (genus), *Microbacteriaceae* (family to genus), *Mycetocola* (genus), *Thermopolyspora* (genus), *Salinispora* (genus), *Luedemannella* (genus). The relative abundances of the family *Thermoactinomycetaceae* and genus *Thermobacillus* were dramatically higher in the b1 treatment than in the other six treatments. In comparison, the family *FFCH13075* and genus *Anaerolinea* were significantly higher in the a3 treatment. Similarly, *Nocardioidaceae* (family), *Polyangiaceae* (family), and *Methylocaldum* (genus) were enriched in the a2 treatment (Fig 3B).

## Relationships between environmental factors and bacterial communities

Applications of organic fertilizer can change microbial community structures and environmental characteristics. Therefore, this study investigated whether microbial community structure and environmental characteristics are related. After the removal of redundant variables, 12 environmental characteristics were chosen for the RDA (Fig 4, S2 Table). The results showed that TP (P = 0.032) and soil urease (P = 0.044) significantly affected the bacterial community structure.

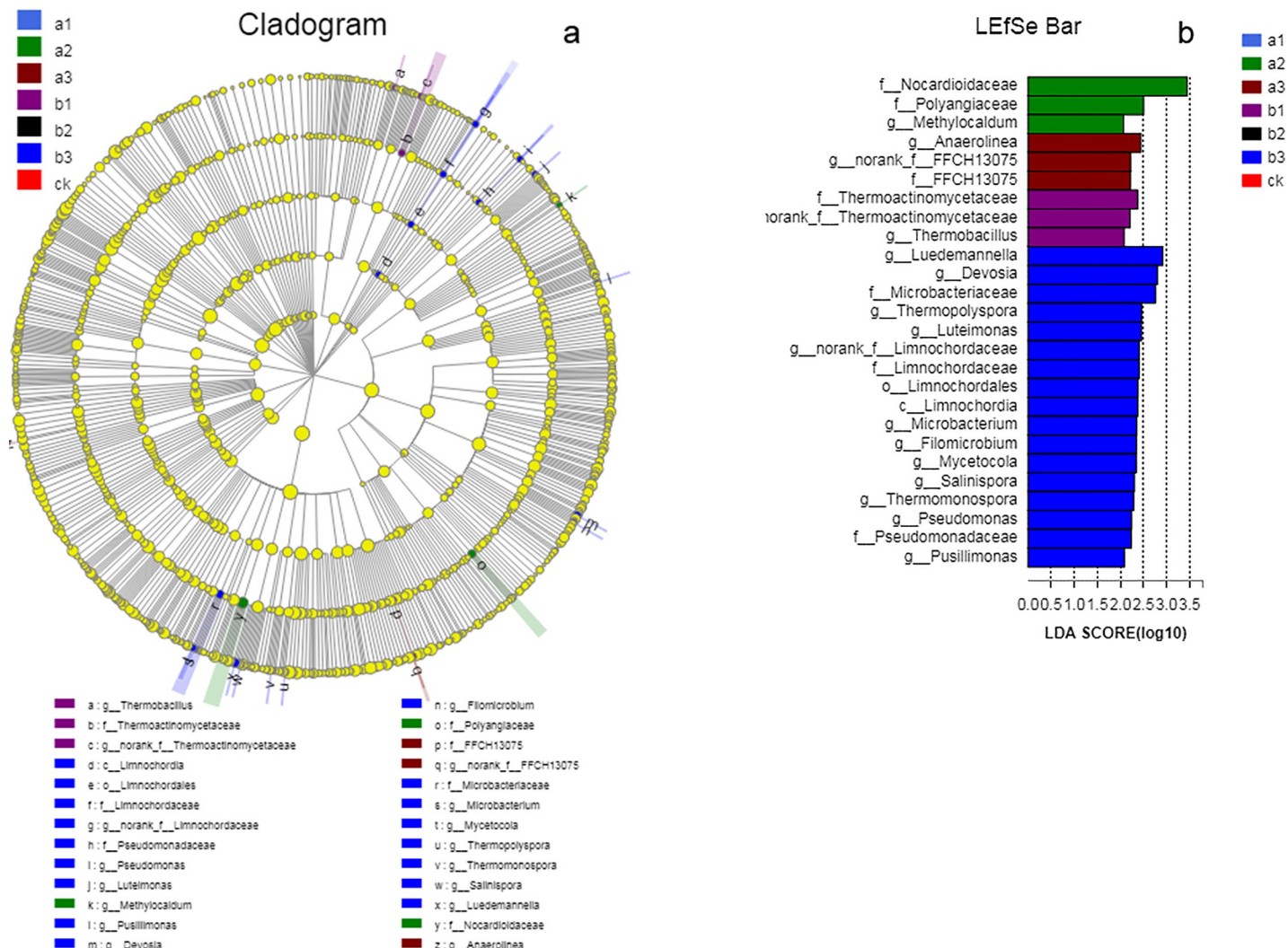

**Fig 3. Cladogram(a) and linear discriminant analysis(b) from the different organic fertilizers treatment.** Cladogram showing the phylogenetic distribution of the bacterial lineages from the different organic fertilizers (a). Differences are represented by the color of the most abundant class (light blue indicates a1; green indicates a2; brick red indicates a3; purple indicates b1; black indicates b2; dark blue indicates b3; red indicates ck). Each circle's diameter reflects the abundance of that taxa in the community. Circles represent phylogenetic levels from the phylum to genus level. The linear discriminant analysis effect size (LEfSe) of bacterial communities with LDA scores higher than 3.0 (b). ck, no fertilizer application treatment; a1, a2, a3, vermicompost fertilizer treatments; b1, b2, b3, mushroom residue fertilizer treatments.

The spearman correlation heatmap showed that soil urease was significantly positively correlated with *Chloroflexi* and *Armatimonadetes*, and had an extremely significantly positive correlation with *Cyanobacteria*. *Fibrobacteres*, *GAL15*, *Latescibacteria*, and *Saccharibacteria* were significantly positively correlated with SOM and DM. *Chlamydiae* were significantly negatively correlated with TK and AN. These results indicate that different bacterial phylum was affected to different extent by soil chemical properties and enzyme activity (Fig 5).

## Discussion

The results demonstrated significant changes in the soil indexes and aboveground grass yield resulting from different organic fertilizer treatments (Table 1). Following the application of organic manure, Chu et al [32] found that SOC and the major soil nutrients (N, P, and K) were also significantly increased in a sandy loam soil. Zhen et al. [33] experiment results

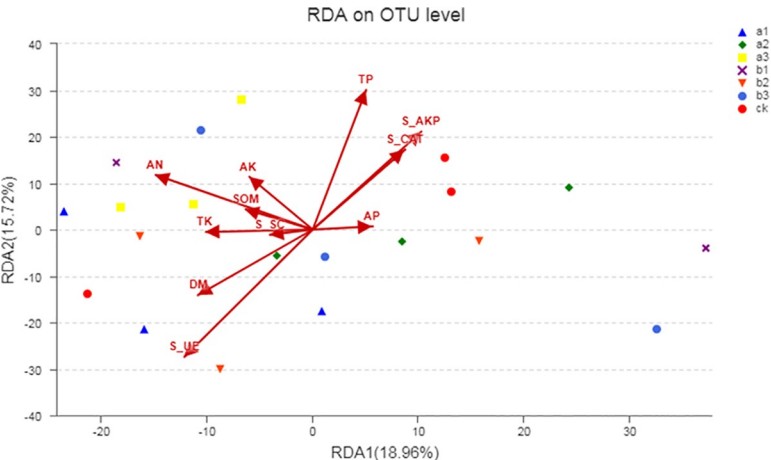

**Fig 4. Redundancy Analysis (RDA).** Based on the relative abundance of bacterial at the operational taxonomic unit (OTU) level and selected soil chemical properties among different organic fertilizer treatments. Note: ck, no fertilizer application treatment; a1, a2, a3, vermicompost fertilizer treatments; b1, b2, b3, mushroom residue fertilizer treatments. TP, total phosphorus; TK, total potassium; AN, available nitrogen; AP, available phosphorus; AK, available potassium; SOM, soil organic matter. S_UE, soil urease; S_SC, soil sucrase; S_AKP, soil alkaline phosphatase; S_CAT, soil catalase.

showed that long-term fertilization influenced soil properties and microbial community. These changes in soil properties can be attributed to the variation in soil microbial community after fertilizer application. The experiments reported here yielded similar results. Organic fertilizer application increased the soil nutrient status to different degrees, with the application of vermicompost and mushroom residues effectively improving the AP and AK content (Table 1). This is in accordance with most other fertilizer application experiments in various ecosystems [20, 27, 34] Soil pH, total nutrients, and organic matter showed no significant changes in response to the fertilizer treatments, suggesting that changes in these indexes were not easily detected in the short term. Similar results were found by Chen et al [35]. Moreover, the mushroom residues, especially the b3 treatment, significantly increased the total biomass of grass compared to other treatments. This is consistent with the view that grass yield is significantly influenced by organic fertilizer application compared with unfertilized controls [36]. These results indicated that the application of vermicompost and mushroom residue organic fertilizers increased the pool of organic materials and nutrient availability in the soil and further improved the physical environment of the soil and the yield of pasture [25].

Soil enzymes are a kind of special protein, with their own biochemical and catalytic properties. They are involved in many important biochemical processes in soil and have a close relationship with soil fertility [37, 38]. The results indicated that the activities of four enzymes did not significantly respond to the different vermicompost and mushroom residue treatments, but soil urease activity was slightly reduced (Table 2). Wang et al. reported similar results [39]. This may be because organic fertilizers are slow-acting fertilizers that release nutrients slowly, while plants absorb most of their nutrients in the growing season. Another reason could be that enzymes are specific and many fertilizers require specific enzymes for an enzymatic reaction to proceed [40]. In addition, different pasture and growing season climatic conditions are important factors in soil enzyme activity. This may affect soil enzyme activity through changes in soil microbial and root exudates and their interactions, which needs to be studied further.

The biodiversity and richness of the microbial community are considered to be critical to the integrity, function, and long-term sustainability of soil ecosystems, but they are usually

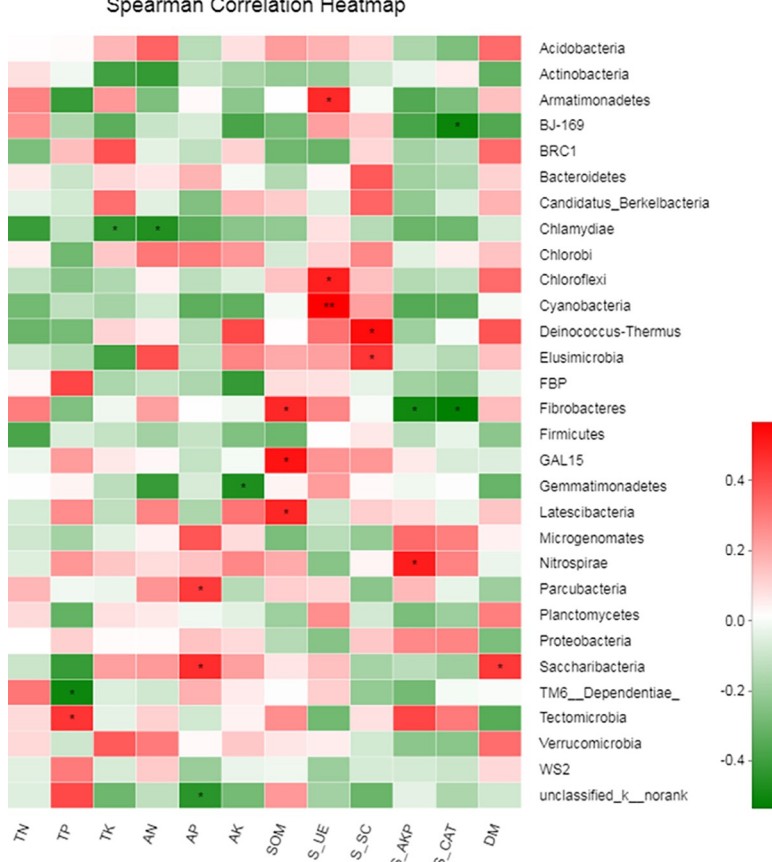

**Fig 5. Spearman correlation heatmap based on the bacterial community and environmental variables.** The X and Y axes of the thermal graph are the environmental factors and species (at phylum level), respectively, and the R and P values were calculated. R values are shown in different colors in the graph. The color scale on the right shows the color partitioning of the different R values. * and ** indicate that means were significantly different at P < 0.05 and P < 0.01, respectively.

changed by agricultural perturbations [41]. For each environment, there likely is an optimum soil microbial community that promotes plant growth and protection from disease [42]. This was confirmed by Hartman et al study [43]. The results of experiment showed that the application of vermicompost (a1) and mushroom residue (b2) organic fertilizers resulted in a significantly increased soil bacterial richness index (Ace) (Table 2). These results may be due to the effects of organic fertilizer application on soil physical and chemical properties and biological characteristics, especially soil pH and the microbial community. In these experiments, most of the raw materials for the vermicompost were obtained from the digestion of livestock and poultry manure. This is generally alkaline, and therefore the pH of the soil tended to be neutral after being applied to the soil. The mushroom residues were mainly the substrate collected after planting mushrooms. A large amount of fungal mycelium remained in the substrate, which was rich in amino acids, cellulose, hydrocarbons, and trace elements. Therefore, the application of vermicompost and mushroom residue organic fertilizers to the soil significantly changed the microbial population and bacterial abundance, which in turn affected the microbial community structure.

Any changes in environmental factors could alter the soil microbial community structure to some extent [44]. Based on the experimental results, the PLS-DA score map clearly distinguished the differences between soil bacterial microbial communities treated with the different

organic fertilizers (Fig 1). These results were consistent with the observations by many long-term fertilizer application studies [45, 46]. This result further indicated that different fertilizers differentially modulate the soil microbial community structure [47, 48]. This indicated that the application of different vermicompost and mushroom residue organic fertilizers affected the soil microbial community structure to varying degrees. The effects of different levels of vermicompost and mushroom residue organic fertilizer on the soil microbial community were different. To understand the relationship between microbial community structure, treatment, and environment, RDA was used (S2 Table). RDA shows that TP and soil urease are important factors affecting the distribution of soil bacterial communities.

Although the bacterial community responded to the different organic fertilizers in different ways with regard to relative abundance and diversity, our study also showed that the soil microbial community structure changed significantly along a gradient of different levels of vermicompost and mushroom residue applications, which was consistent with the changes in soil and plant characteristics. An analysis of phylum abundance indicated that *Actinobacteria*, *Proteobacteria*, *Acidobacteria*, *Verrucomicrobia*, *Chlorofexi*, and *Gemmatimonadtes* were the six most dominant phyla across all treatments (Fig 2). These results are in agreement with previous studies on bacterial community composition in grassland soils [25, 49–51]. Members of *Actinobacteria* are most common in grassland soils and are very suitable for this environment. Actinobacteria play a certain role in the natural nitrogen cycle, thereby accelerating the soil nutrient cycle, which is conducive to the growth of plants and the formation of good soil nutrient conditions. In addition, *Actinobacteria* play a key role in the decomposition of organic matter and the humus formation process [52, 53], which can produce a variety of antibiotics to protect soil and plant roots from pathogenic microorganisms [54]. A similar abundance among fertilizer treatments was reported by Chen et al. based on a long-term fertilizer application trial [55]. *Proteobacteria* are extremely rich in species and genetic diversity, which determines that this group covers a very wide range of physiological metabolism types. These include many bacteria that can fix N and a variety of metabolic species, many of which can use photosynthesis to store energy. Therefore, the species within the phylum *Proteobacteria* have important applications in the fields of agriculture, industry, medicine, health, environmental protection, etc., and are widely used in the promotion of nitrogen fertilizer utilization, plant disease and insect pest control, industrial and agricultural wastewater treatment, soil remediation, and complex pollutant degradation. In addition, these organisms increase the utilization of phosphate through dissolution, which may also benefit plant growth [56, 57]. *Acidobacteria* is a newly isolated bacterium that has been the focus of little research, but they play an important role in the soil ecosystem. *Verrucomicrobia* is mainly found in aquatic and soil environments or human feces. Some studies indicated that *Verrucomicrobia*, which is involved in organic C utilization, was more abundant following long-term large inputs of chemical fertilizer than in soils with lower or no chemical fertilizer application [58]. It can enhance the photosynthetic efficiency of the leaves and greatly enhance the accumulation of pasture materials. *Chlorofexi* contains green pigment, and its presence is significantly positively correlated with water holding capacity and aboveground biomass. Some members of this family are facultative anaerobes and can degrade cellulose [59, 60]. The most abundant families, *Nocardioidaceae* and *Polyangiaceae*, and the *Methylocaldum* genus were observed in the a2 treatment (Fig 3), which produced antibiotics and degraded cellulose [61]. The *Anaerolinea* genus and *FFCH13075* family were the most abundant under the a3 treatment. *Anaerolinea* produces organic nutrients, fermenting organic acids, protein extracts, and a few carbohydrates. Its presence shows that the application of a vermicompost organic fertilizer affected some of the activities of soil microbes, which in turn affected soil conditions and plant growth. After applying different levels of mushroom residue organic fertilizer, with the exception of the enrichment of

the *Thermoactinomycetaceae* family and *Thermobacillus* genus in b1, the LEfSe analysis identified that the biological flora were most abundant in the b3 treatment (Fig 3). These microbial clades (*Devosia*, *Microbacteriaceae*, *Thermopolyspora*, and *Salinispora*) all have a close relationship with cellulose degradation, respiratory metabolism, and carbohydrate synthesis [62]. *Pseudomonadaceae* has the ability to break down proteins and fats, rapidly decomposing humus in the soil to provide nutrients for plants [63]. Our results indicated that nutrient acquisition by the application of organic fertilizer would concomitantly promote the decomposition of organic materials and affect the soil microbial flora. Jangid et al [64] also showed that agricultural management practices, especially fertilizer treatments, over a long time, significantly alter the soil microbial community.

In addition, we found that the environmental changes that occur with the application of organic fertilizers to soil contributed differently to the different microbial groups in the community (Fig 4). Many studies have also shown that soil environmental factors affect the microbial community structure [41, 65, 66]. For example, Wei et al. found that soil SOC and TN were the main drivers of microbial community structure based on 35 years of manure and chemical fertilizer applications [67]. However, Wang et al. indicated that soil pH, organic matter, and AP concentrations were the important factors in shaping bacterial communities in the maize rhizosphere [54]. In this study, the RDA revealed that TP and soil urease were the key factors driving the distribution and composition of the soil bacterial community (Fig 4, S2 Table). The spearman correlation heatmap showed that environmental factors were significantly correlated with different microbial flora (Fig 5). Urease plays an important role in the C and N cycles in soil ecosystems and is positively correlated with soil microbial biomass, organic matter content, and N content. For example, EO and Park [68] showed that N has significant effects on the bacterial community. The urease-related *chloroflexi* provides energy for above-ground plants through photosynthesis and contains a green pigment. *Fibrobacteres* is associated with organic matter following the breakdown of cellulose [69, 70]. The application of organic fertilizers to environmental factors and the microbial community requires long-term experimental studies, because it is often difficult to determine the relationship between microbial communities and soil nutrient cycling [71]. These experimental results only indicate short-term fertilizer application effects. The long-term mechanism and the duration of the effect of organic fertilizers on soil microorganisms have yet to be monitored over the long-term.

## Conclusions

The application of different organic fertilizers formed different bacterial community structures in the soil of a L. chinensis grassland. High levels of vermicompost and mushroom residues not only increased the availability of organic matter reservoirs and nutrients, but also enhanced the biodiversity of soil bacterial communities in a L. chinensis grassland. It also increased the levels of effective P, K, and biomass, and the abundance of Actinobacteria. The results indicated that soil fertility and quality improved after fertilization. Based on these results, the application of high levels of vermicompost and mushroom residues is a practical option for enhancing soil available nutrient and bacterial community diversity in a L. chinensis grassland. However, the specific mechanisms by which organic fertilizer impacts on available nutrients and flora in soils needs further research.

## Supporting information

**S1 Fig. Rarefaction curves for each sample.** ck, no fertilizer application treatment; a1, a2, a3, vermicompost fertilizer treatments; b1, b2, b3, mushroom residue fertilizer treatments.
(TIF)

**S1 Table. Sequencing data and OTUs across treatments.**
(DOCX)

**S2 Table. Correlation between 16s rRNA data of bacteria treated with organic fertilizer and environmental factors.** The results of a permutation test of environmental variables and soil samples based on the bacterial 16s rRNA data at the operational taxonomic unit (OTU) level. The p values were based on 999 permutations.
(DOCX)

## Acknowledgments

The authors would like to thank the staff of Hulunbeier test station. The authors are grateful to Zongyong Tong for their kind help in conducting the experiment.

## Author Contributions

**Conceptualization:** Lirong Shang, Liqiang Wan, Xianglin Li.

**Data curation:** Lirong Shang.

**Formal analysis:** Lirong Shang.

**Investigation:** Lirong Shang, Liqiang Wan, Xiaoxin Zhou, Shuo Li, Xianglin Li.

**Methodology:** Lirong Shang, Xianglin Li.

**Resources:** Liqiang Wan, Xianglin Li.

**Writing – original draft:** Lirong Shang.

**Writing – review & editing:** Lirong Shang, Liqiang Wan.

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
