## [Decision Letter · Decision Letter 0]

21 Aug 2020

PONE-D-20-23186

Effects of organic fertilizer on soil nutrient status, enzyme activity, and bacterial community diversity in Leymus chinensis steppe in Inner Mongolia, China

PLOS ONE

Dear Dr. Wan 

Thank you for submitting your manuscript to PLOS ONE. After careful consideration, we feel that it has merit but does not fully meet PLOS ONE’s publication criteria as it currently stands. Therefore, we invite you to submit a revised version of the manuscript that addresses the points raised during the review process.

We look forward to receiving your revised manuscript.

Kind regards,

Tunira Bhadauria, Ph.D.

Academic Editor

PLOS ONE

Additional Editor Comments:

1)The manuscript is well written and well presented but there are some comments made by the reviewers whih needs to be incorporated in the text.

for example replacing ter fertilization by fertilizer application.Attention needs to be paid to references highlighted in the text

2)Discussion should be elaborated so as to make the manuscript more acceptable to the readers.

3)In result section please mention only significant results

4)Other comments are mentioned in the text

5) The manuscript can be aepted for publication only after major revision.

Reviewers' comments:

Reviewer's Responses to Questions

**Comments to the Author**

1. Is the manuscript technically sound, and do the data support the conclusions?

Reviewer #1: Yes

Reviewer #2: Yes

2. Has the statistical analysis been performed appropriately and rigorously? 

Reviewer #1: Yes

Reviewer #2: Yes

3. Have the authors made all data underlying the findings in their manuscript fully available?

Reviewer #1: Yes

Reviewer #2: Yes

4. Is the manuscript presented in an intelligible fashion and written in standard English?

Reviewer #1: Yes

Reviewer #2: Yes

5. Review Comments to the Author

Reviewer #1: Comments:

1) Line no 24: “that” is repeated

2) Line no 55: put a full stop after “processes”

3) Line no 56: use the term “microbial” instead of “microbe”

4) Line no 55-57: “Because of……..after fertilization”, provide some references of similar studies

5) Line no 60: Instead of “studies of the” use “studies on the”

6) Line no 68-70: use either of the two terms “were conducted” or “was conducted”

7) Line no 193: use another term in place of “dramatically”

8) Line no 356: mention reference number for Chu et al.

9) Line no 195-197: The percentage increase in soil parameters viz. TP and TK mentioned in the result section is not significant as compared to control values. Only significant results should be mentioned in the result section supported by appropriate references in the discussion part.

10) Line no 199-200: The percentage increase in soil parameters viz. AN and SOM is again not significant in comparison to control values.

11) Line no 201-202: In addition to mentioned parameters (TN, AN & SOM), TP, TK and AP (except for treatment a3) also didn’t show significant variations and should be mentioned.

12) Line no 456-457-458: Provide some references.

13) Provide some recent references of current year.

14) The discussion section can be further strengthened by citing more similar studies.

Reviewer #2: The research work has been done in intelligent manner. Data has been put with sound statistical analysis. However certain point need to be corrected as:

1.The term 'fertilization' often used in manuscript which should be replaced with 'fertilizer application'

2.few reference nos are not correct which are highlighted/corrected

3.Discussions particularly related to sequencing of microbial community diversity need more clear presentation.

6. PLOS authors have the option to publish the peer review history of their article (what does this mean?). If published, this will include your full peer review and any attached files.

Reviewer #1: No

Reviewer #2: No

---

## [Author Response · Author response to Decision Letter 0]

24 Sep 2020

Response to Reviewer 1 Comments

Reviewer #1: Comments:

1) Line no 24: “that” is repeated

Response 1): We thank the reviewer for the time in closely viewing our manuscript. Done. 

2) Line no 55: put a full stop after “processes”

Response 2): It has been modified in the revised version..

3) Line no 56: use the term “microbial” instead of “microbe”

Response 3): Done. The sentence has been revised as suggested by the reviewer. 

4) Line no 55-57: “Because of……..after fertilization”, provide some references of similar studies

Response 4): The reviewer’s suggestion has been adopted and some similar references have been cited in the revision. 

5) Line no 60: Instead of “studies of the” use “studies on the”

Response 5): It has been modified in the revised version. 

6) Line no 68-70: use either of the two terms “were conducted” or “was conducted”

Response 6): Modified in the revised version. Use “was conducted”. 

7) Line no 193: use another term in place of “dramatically”

Response 7): The reviewer’s suggestions have been adopted. It has been modified in the revised version. Use “significantly”. The application of vermicompost and mushroom residue organic fertilizers significantly altered soil chemical properties (Table 1).

8) Line no 356: mention reference number for Chu et al.

Response 8): Added chu et al references in the revised version. 

9) Line no 195-197: The percentage increase in soil parameters viz. TP and TK mentioned in the result section is not significant as compared to control values. Only significant results should be mentioned in the result section supported by appropriate references in the discussion part.

Response 9): The reviewer’s suggestion has been adopted and only significant results should be mentioned in the result section supported by appropriate references in the discussion part. Corresponding references have been added to the discussion section of the revised version. Following the application of organic manure, Chu et al [32] found that SOC and the major soil nutrients (N, P, and K) were also significantly increased in a sandy loam soil. Zhen et al. [33] experiment results showed that long-term fertilization influenced soil properties and microbial community. These changes in soil properties can be attributed to the variation in soil microbial community after fertilizer application.

10) Line no 199-200: The percentage increase in soil parameters viz. AN and SOM is again not significant in comparison to control values.

Response 10): The reviewer’s suggestion has been adopted and this part has been modified in the revised version. Soil AN and SOM were not significantly affected by vermicompost and mushroom residue applications.

11) Line no 201-202: In addition to mentioned parameters (TN, AN & SOM), TP, TK and AP (except for treatment a3) also didn’t show significant variations and should be mentioned.

Response 11): The reviewer’s suggestion has been adopted and this part has been modified in the revised version.

12) Line no 456-457-458: Provide some references.

Response 12): It has been modified in the revised version and added corresponding references.

13) Provide some recent references of current year.

Response 13): The reviewer’s suggestion has been adopted and an updated reference has been cited in the revision.

14) The discussion section can be further strengthened by citing more similar studies.

Response 14): The reviewer’s suggestions have been adopted and the discussion section has been revised, and corresponding similar documents have been added. 

Response to Reviewer 2 Comments

Reviewer #2: The research work has been done in intelligent manner. Data has been put with sound statistical analysis. However certain point need to be corrected as:

1. The term 'fertilization' often used in manuscript which should be replaced with 'fertilizer application'

Response 1: We thank the reviewer for the time in closely viewing our manuscript. The "fertilization" in the article has been changed to "fertilizer application".

2. few reference nos are not correct which are highlighted/corrected

Response 2: Done. The references of the manuscript have been checked and revised in the revised version.

3. Discussions particularly related to sequencing of microbial community diversity need more clear presentation.

Response 3: The reviewer’s suggestion has been adopted. We have revised the manuscript based on the reviewer’s comments and discussion related to sequencing of microbial community diversity has been modified. 

Response to Editor Comments

1) The manuscript is well written and well presented but there are some comments made by the reviewers whih needs to be incorporated in the text.

for example replacing ter fertilization by fertilizer application. Attention needs to be paid to references highlighted in the text

Response 1）: We thank the editor for affirming the manuscript. The "fertilization" in the manuscript has been changed to "fertilizer application". The references of the manuscript have been checked and revised in the revised version.

2) Discussion should be elaborated so as to make the manuscript more acceptable to the readers.

Response 2): The editor’s suggestion has been adopted. We revised the discussion part in the revised version.

3) In result section please mention only significant results

Response 3): Done. We revised the results section in the revised version and only mentioned important results.

4) Other comments are mentioned in the text

Response 4): Done. We have completed the revision of the full text in the revised version.

---

## [Editor Report · Decision Letter 1]

29 Sep 2020

Effects of organic fertilizer on soil nutrient status, enzyme activity, and bacterial community diversity in Leymus chinensis steppe in Inner Mongolia, China

PONE-D-20-23186R1

Dear Dr. WAN

We’re pleased to inform you that your manuscript has been judged scientifically suitable for publication and will be formally accepted for publication once it meets all outstanding technical requirements.

Kind regards,

Tunira Bhadauria, Ph.D.

Academic Editor

PLOS ONE

A

---

## [Editor Report · Acceptance letter]

2 Oct 2020

PONE-D-20-23186R1 

Effects of organic fertilizer on soil nutrient status, enzyme activity, and bacterial community diversity in *Leymus chinensis* steppe in Inner Mongolia, China 

Dear Dr. Wan:

I'm pleased to inform you that your manuscript has been deemed suitable for publication in PLOS ONE. Congratulations! Your manuscript is now with our production department. 

Kind regards, 

on behalf of

Dr. Tunira Bhadauria 

Academic Editor

PLOS ONE